# Seawater-resistant emulsified epoxy resin for effective sand control in unconsolidated sandstone oil reservoir

Chenyang Tang[1], Wei Zheng[1], Yufei He[1], Shaobing Cai[1], Juanzhe Jiang[1], Yue Pan[1], Ke Li[1], Xiaoxia Shang [2,3]*

1 CNOOC Research Institute Co., Ltd., Beijing, China, 2 School of Petroleum Engineering, Yangtze University, National Engineering Research Center for Oil & Gas Drilling and Completion Technology, Wuhan, China, 3 Hubei Key Laboratory of Oil and Gas Drilling and Production Engineering, Yangtze University, Wuhan, China

* ShangXX27@163.com

## Abstract

Sand production in oil wells is recognized as a persistent challenge during oilfield development, adversely affecting well productivity and operational stability. Chemical sand control methods, particularly resin-based sand consolidation, are considered a promising solution due to their operational simplicity and effectiveness. However, conventional emulsified resins are known to be highly sensitive to high-salinity environments, which can lead to emulsion destabilization and reduced consolidation strength. To address this limitation, a novel emulsified epoxy resin system was developed in this study using a nonionic emulsifying curing agent—fatty amine poly(epoxy ethyl ether)—by which salinity tolerance is significantly enhanced, supporting dilution water salinity up to $3.8 \times 10^4$ mg/L. Through single-factor experiments, an optimal formulation was identified as 16% epoxy resin, 24% emulsified curing agent, 1% coupling agent, and 5.6% stabilizer. The molecular structure of the emulsified resin and the stability of the cured matrix were thoroughly characterized. The effects of curing temperature, time, sand particle size, and stabilizer dosage on compressive strength and permeability were systematically evaluated. It was demonstrated that after being cured at 80 °C for 12 hours, the consolidated cores achieved a compressive strength exceeding 3 MPa with permeability retention above 75%. Furthermore, the consolidated cores were shown to exhibit excellent long-term stability, maintaining their mechanical and flow properties after 30-day immersion in kerosene, 10% HCl, and formation water. This study bridges a critical research gap in high-salinity applications of water-based resin emulsions and provides a robust technical solution for sand control in challenging reservoir environments.

**Data availability statement:** All relevant data for this study are publicly available from the Mendeley Data repositories (https://doi.org/10.17632/zxrd8gmsxw.1) (https://doi.org/10.17632/xwf77hxf4s.1).

**Funding:** This work was supported by the National Natural Science Foundation of China (Grant Number 52374028、W2421011). The funders had no role in the study design, data collection and analysis, decision to publish, or preparation of the manuscript.

**Competing interests:** The authors have declared that no competing interests exist.

## Introduction

Sand production is widely recognized as a pervasive challenge during oilfield development, exerting direct or indirect adverse effects on well productivity and operational stability. Consequently, the development and application of effective sand control technologies are urgently required to address these challenges, thereby ensuring high and sustainable oilfield production. With advancements in sand control technology, two primary categories have been established: mechanical sand control and chemical sand control. Mechanical sand control can be further classified into two types. The first type involves the deployment of sand control screens, such as slotted liners, wire-wrapped screens, cemented sand filter pipes, or double-layer/multi-layer sand pipes [1]. Although this technology is considered straightforward and easily implementable, its effectiveness is often limited, and its service life is relatively short. The main issue is that the gaps or pores in the sand control screens are easily clogged by fine sand entering the wellbore. The second method involves gravel packing after the sand control screen is installed. This approach effectively confines sand production from the formation and maintains a stable mechanical structure within the reservoir. It is recognized as an effective sand control solution with an extended service life [2–5]. However, this method is associated with high installation costs. In high-salinity environments, metal sand control pipes may undergo corrosion and become clogged, and acidification treatments can further damage the sand control equipment.

Compared to mechanical methods, chemical sand control technology is widely employed in oil and gas wells due to its operational simplicity and ease of subsequent processing. Chemical sand control is a sand consolidation technique wherein specific chemical agents and solid particles (e.g., pre-packed gravel) are injected into the formation around the wellbore and perforation zones. This process consolidates the formation sand, thereby reducing sand production and enabling sustainable long-term production. The most significant advantage of chemical sand control is that no mechanical equipment is left in the wellbore, and the construction process is simplified, requiring only the injection of chemical agents or a mixture of chemical agents and proppants.

This method is particularly suitable for thin layers with relatively uniform permeability and demonstrates superior performance in silty sandstone formations compared to mechanical sand control. It is also recommended for sand control in the upper layers of dual completion operations. Chemical sand control can be broadly categorized into three types: chemical sand consolidation [6,7], artificial barrier sand control, and other chemical sand consolidation methods [8]. A common chemical sand consolidation method involves the injection of organic or inorganic chemical agents into unconsolidated oil-bearing layers or pre-packed sand layers, resulting in the cementation of sand grains into a consolidated layer with certain strength and favorable permeability [9–13]. Currently, chemical agents used for sand consolidation include film-forming sand control agents [14], oil-based resin sand control agents, water-based resin sand control agents, nano-fluid sand control agents, and polymer sand control agents. Each method possesses distinct advantages and disadvantages, which are summarized in Table 1.

**Table 1. Summary of chemical sand fixer characteristics.**

| Sand Consolidation System | Advantages | Disadvantages |
|---|---|---|
| Film-forming sand control agents | Minimal formation damage, high permeability, effective in-situ consolidation of various particles (e.g., clay, fine sand, and packed sand). | High cost and limited by technological patents. |
| Oil-based resin sand control agent | Broad temperature adaptability, excellent chemical and moisture resistance, cost-effective, with strong sand-consolidation capability. | Potential for significant formation damage; difficult to remediate if plugging occurs. |
| Water-based resin sand control agent | Compatible with injection systems, low viscosity for easy injection, easy cleanup, minimal formation damage, high consolidation strength, and economical. | Difficulty in simultaneously achieving high consolidation strength and high permeability retention. |
| Nano-fluid sand control agents | Low formation damage, high permeability, recyclable after consolidation failure, effective across a wide range of sand grain sizes. | High cost and requires specific nanoparticle carrier fluids. |
| Polymer sand control agents | Extensive coverage area, high consolidation strength (via physico-chemical mechanisms), excellent temperature and salt tolerance. | Complex formulation process and high cost. |

Among them, resin cemented sand is the most widely used, and resin cementing agents mainly contain phenolic resins, epoxy resins [15], furan resins [16], urea-formaldehyde resins [17], and so on. Modified the structure of phenolic resin based on its curing mechanism, enabling the material to achieve higher consolidation strength at lower temperatures. Nguyen et al. [18] proposed a feasible method for stabilizing the near-wellbore region using a low-viscosity, single-component curing resin system. Their results demonstrated that the resin treatment significantly enhanced the consolidation strength of unconsolidated and weakly consolidated formations. Keith et al. [19] applied furan resin for sand consolidation in oilfields on the Malaysian Peninsula, which resulted in a substantial increase in oil production post-treatment. Tabbakhzadeh et al [20] employed both epoxy and furan resins for sand consolidation; the compressive strength of epoxy-consolidated cores reached up to 32.82 MPa. Dewprashad et al. [21] developed a novel epoxy resin system that exhibited exceptional compressive strength under high-temperature conditions. Gu et al. [22] prepared a new sand consolidation agent by blending epoxy and phenolic resins. Their experiments showed that the agent could effectively consolidate sand in a 50 °C water bath, yielding a compressive strength of approximately 5 MPa and an equivalent liquid permeability of about 1.2 μm². Shang et al. [23] investigated a new foamed amino resin system for oilfield sand control. Cured at 60 °C for about 12 hours, the consolidated cores achieved a compressive strength of 6.28 MPa. Talaghat et al. [24] mixed modified phenolic resin with sand samples from Ahwaz and Mansoori oilfields at varying ratios to prepare consolidated cores. These samples exhibited permeabilities ranging from 1500 to 3500 mD, porosities between 38% and 68%, and compressive strengths exceeding 20 MPa. Chang et al. [25] studied a new polymer resin formulation incorporating a low-viscosity epoxy resin emulsion to enhance adhesion between sand particles. The treated unconsolidated sand showed effective consolidation, with a compressive strength greater than 6 MPa and no sand production observed in the effluent even under high flow rates.

However, a critical limitation persists in prior studies and existing technologies: their formulations have been predominantly optimized and validated in freshwater or low-salinity environments (typically below 15,000 mg/L). Under high-salinity conditions—such as those found in offshore reservoirs or formations with elevated-salinity formation brines—the performance and stability of conventional emulsified resins are significantly compromised due to electrolyte-induced emulsion breakdown. This issue represents a major research gap and a practical challenge for applying chemical sand consolidation methods in such demanding settings. The central question this study seeks to address is therefore: How can a stable and effective aqueous-based epoxy resin emulsion system be formulated to achieve robust sand consolidation performance under high-salinity conditions (up to 38,000 mg/L)? To bridge this gap, a novel emulsion system was engineered using a nonionic emulsifying curing agent, specifically fatty amine poly(ethylene oxide) ether, designed to resist salinity-induced destabilization. The formulation was systematically optimized through rigorous experimental design, and its performance was comprehensively evaluated to establish a viable solution for high-salinity applications.

## Materials and methods

### Instruments and reagents

**Main reagents.** Epoxy resin (E-51) was supplied by Kunshan Jiulimei Electronic Materials Co., Ltd. Fatty amine polyoxyethylene ether (industrial grade) was obtained from Tianjin Zhonghe Shengtai Chemical Co., Ltd. KH550 (industrial grade) was provided by Shandong Hengyu New Material Co., Ltd. The Control Agent (analytical grade) was sourced from Sinopharm Group Chemical Reagent Co., Ltd. Aromatic amine polyoxyethylene ether (industrial grade) was supplied by Guangzhou Qihua Chemical Co., Ltd. Polyacid amine polyamine dispersion (industrial grade) was obtained from Guangdong Wengjiang Chemical Reagent Co., Ltd.

**Main instruments.** Thermogravimetric analysis was performed using a TGA-Q500 analyzer (TA Instruments, USA). Viscosity measurements were carried out with a DVNext viscometer (Brookfield, USA). Microstructural observation was conducted on an RX50M metallographic microscope (YuBO Shunyu Instrument Co., Ltd., China). Mechanical properties were tested with a WDW-50 universal testing machine (Jinan West Machinery Equipment Co., Ltd., China). Morphological characterization was achieved using a Quanta 450 scanning electron microscope (FEI Corporation, USA). Core flooding experiments were performed with a core displacement device (Jiangsu Lianyou Scientific Research Instrument Co., Ltd., China).

### Experimental methods

**Characterization tests. Thermogravimetric analysis:** The thermal stability of the emulsion consolidation system was evaluated using a TGA-Q500 thermogravimetric analyzer (TA Instruments, USA) over a temperature range of 40–600°C.

**Test experiments. Preparation method of consolidation system:** A predetermined amount of epoxy resin, emulsifying curing agent, and coupling agent was homogeneously mixed to form a resin adhesive. The resulting emulsified resin was then thoroughly blended with quartz sand to produce a pre-cured sand mixture. This mixture was packed into a glass tube sealed at one end with a rubber plug. The tube was subsequently shaken and compacted to achieve a leveled sand surface. Curing was carried out by placing the tube in an oven at 80 °C for 3 days. After curing, the sample was removed and allowed to cool to room temperature. Finally, the compressive strength and permeability of the consolidated sample were measured.

**Viscosity test:** The rheological properties are crucial for evaluating the injectability of the consolidation fluid into the formation. The viscosity of the system was measured using a DV Next rotational viscometer (Brookfield, USA). Measurements were conducted with rotor No. 0 at a rotational speed of 6 rpm. The shear rate was ramped from 1 to 100 $s^{-1}$ to characterize the flow behavior under conditions simulating both near-wellbore (high shear) and deep formation (low shear) flow. The emulsion was prepared for viscosity measurements at various temperatures and for evaluation of its rheological properties.

**Determination of curing time:** The reaction between epoxy resin and the amine curing agent is exothermic. As the reaction proceeds, both the viscosity and temperature of the system increase. The curing time of the emulsion consolidation system can be determined by monitoring the temperature change. The specific experimental procedure was conducted as follows: a specified amount of the emulsion sand-consolidation liquid was placed in a beaker, and an electronic thermometer was inserted to record temperature changes in real time. The beaker was then immersed in a water bath maintained at a predetermined temperature. The curing time was identified as the point at which a sharp increase in both viscosity and temperature was observed. The acceleration of the cross-linking reaction and the transition from a liquid to a gel state are signified by this sharp increase.

**Determination of compressive strength:** The cured consolidated sample was carefully extracted from the glass tube, and both ends were polished to achieve smooth and parallel surfaces. The ends must be polished to ensure uniform load distribution during the compression test and to prevent premature failure caused by stress concentrations at irregular

surfaces.The compressive strength of the consolidated core was determined using a universal mechanical testing machine in accordance with the Chinese National Standard GB/T 50266−2013 (Standard for Test Methods of Engineering Rock Masses). Compressive strength is a key mechanical property indicator used to evaluate the ability of the consolidated sand matrix to withstand stresses and prevent sand production under flowing conditions.

**Displacement experiment:** Rock chips collected from the site were added to the sand-packed tube and compacted. The initial permeability was determined by injecting formation water at a constant rate of 1 mL/min. Pressure back-pressure valves were installed at both ends of the sand-packed tube and set to 3 MPa. Back-pressure is applied to help maintain dissolved gases in solution and to better simulate downhole pressure conditions, leading to more representative permeability measurements. Subsequently, 1.2 pore volumes (PV) of sand consolidation fluid were injected at the same rate (1 mL/min), after which the valves at both ends of the tube were closed. 1.2 PV is injected to ensure complete displacement of the original fluids within the sandpack and sufficient saturation of the pore space with the consolidation fluid, accounting for potential adsorption and bypassing. The tube was then placed in an oven and aged at 80°C for 3 days. After aging, the permeability retention rate and sand production rate of the consolidated sand pack were measured. The permeability retention rate is used to assess the degree of formation damage caused by the consolidation treatment, while the sand production rate is used to evaluate the effectiveness of sand control under flow conditions.

**Micro performance test:** Upon consolidation, the sand column was trimmed and sputter-coated with a gold layer. The consolidated surface was subsequently examined using an FEI Quanta 450 scanning electron microscope (SEM).

## Experimental assumptions and boundary conditions

The experimental investigation and subsequent analysis in this study were conducted under the following explicit assumptions and boundary conditions: Sand Grain Composition: The consolidation experiments primarily assume quartz sand as the representative formation sand. The presence of significant clay minerals or other highly reactive minerals may alter the chemical interaction and consolidation effectiveness. Formation Homogeneity: The laboratory-prepared sand packs assume a relatively homogeneous and unconsolidated formation model. Complex heterogeneities, natural fractures, or pre-existing stresses in real reservoirs are not accounted for. Curing Conditions: The curing process is assumed to be primarily thermally driven at the specified temperatures (e.g., 80°C). The potential influences of downhole pressure on curing kinetics and final properties were not investigated in this phase of study. Fluid Compatibility: The assessment of chemical resistance assumes exposure to the specified fluids (formation water, 10% HCl, kerosene) under static conditions. Dynamic flow conditions or interactions with mixed/complex wellbore fluids are beyond the current scope. Scale: The experiments are conducted on a laboratory scale. Scaling effects for field application, including placement efficiency, radial flow effects, and large-volume mixing, are not considered. These boundaries define the specific context within which the results and conclusions of this study are valid.

## Results and discussion

### Formulation study of the emulsion consolidation system

The formulation was optimized by investigating the content of epoxy resin, emulsified curing agent, coupling agent, and control agent. The influence of each component's dosage on the curing time and compressive strength of the cured composite was evaluated to determine the optimal proportions for practical use.

**Dosage of resin-emulsifying curing agent.** According to the preparation procedure of the emulsion sand consolidation system, the mass fraction of the controlling agent was fixed at 3.6%, and that of the coupling agent at 1%. After adding the resin and emulsified curing agent to form the emulsion, the mixture was injected into pressure-resistant bottles containing 50-, 160-, and 500-mesh quartz sand. The bottles were then placed in an oven and cured at 80 °C for 1–5 days to prepare consolidated cores, after which the compressive strength was measured.

Epoxy resin contains chemical groups such as epoxy groups, hydroxyl groups, ether bonds, and benzene rings (molecular structure shown in Fig 1). These groups impart cured epoxy resin with excellent adhesion, mechanical properties, chemical resistance, and thermal stability [26–28]. The main component of epoxy resin E51 is bisphenol A diglycidyl ether. It has a viscosity of 11,000–14,000 mPa·s at 25 °C, a molecular weight of 370–380, and an epoxy value of 0.49–0.53. Due to its relatively low viscosity and high epoxy value, E51 resin was selected for this curing system. Among three types of curing agents capable of emulsifying epoxy resin—modified polyamine, modified aromatic amine, and baked curing agent—modified polyamine was chosen owing to its high flash point, low viscosity, good compatibility with the resin, and extended curing time [29–31]. Resin emulsions with different ratios were mixed with 160-mesh quartz sand and cured in a water bath at 80 °C for 5 days. The compressive strength of the sand columns was measured for each formulation. As shown in Fig 2, the sand column strength reached its maximum when the mass ratio of emulsifying curing agent to resin was 1.5:1, indicating that this is the optimal ratio.

The effects of dispersed phase mass fraction, quartz sand particle size, and aging time on the strength of the consolidated sand bodies were further analyzed. The results are presented in Figs 3 and 4. To achieve a consolidated sand system with improved mechanical properties, a dispersed phase content of 40% was identified as optimal.

**Control dosage of agent.** The mass fraction of epoxy resin was maintained at 16%, and that of the emulsifying agent at 24%. Variations in temperature led to differences in the curing time of the consolidated sand matrix. Accordingly, the

**Fig 1. Molecular formula of epoxy resin.**

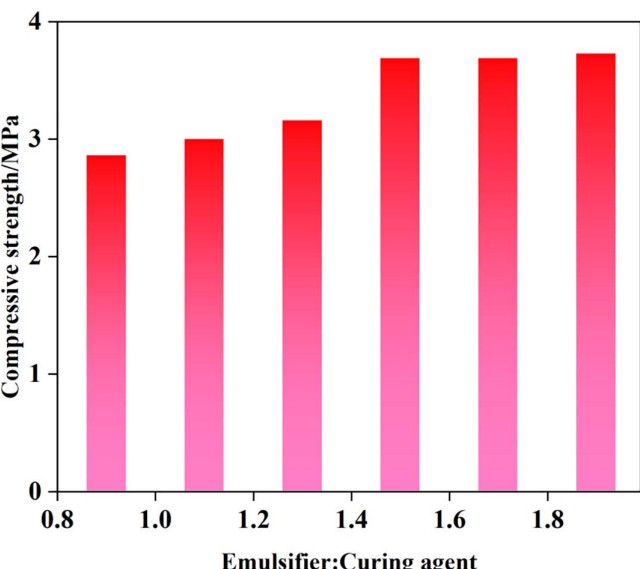

**Fig 2. The compressive strength of sand column under different proportions.**

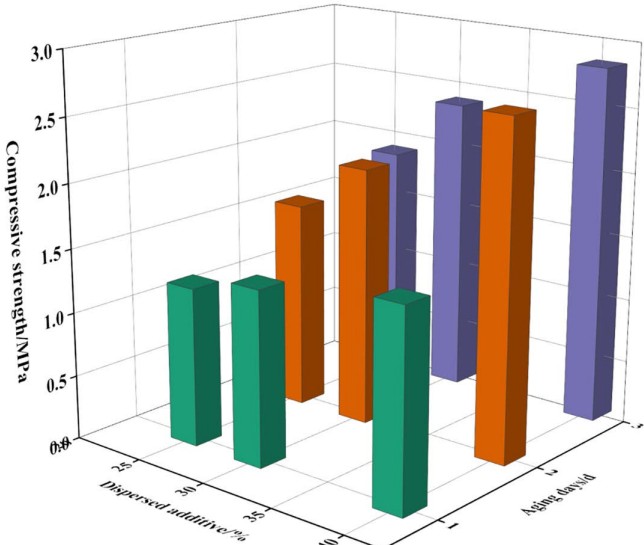

**Fig 3. The effect of dispersed phase addition-aging time on consolidation effect.**

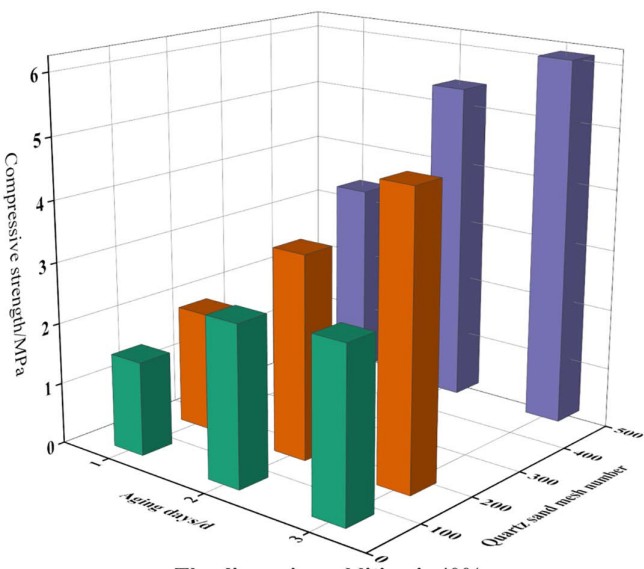

**Fig 4. The effect of quartz sand mesh-aging time on consolidation effect.**

curing behavior of the emulsion was investigated within a temperature range of 50°C to 80°C. The results, presented in Fig 5, demonstrate that the curing time decreased significantly with increasing temperature.

The effect of temperature on the compressive strength of consolidated samples composed of 10% clay and 160-mesh quartz sand was further investigated. As shown in Fig 6, when the control agent was added at a specified concentration, the compressive strength of the consolidated sand body did not increase monotonically with temperature, but rather

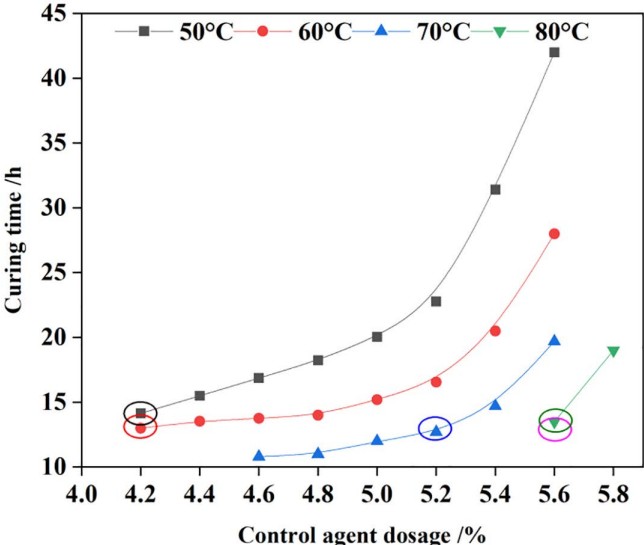

**Fig 5. Effect of temperature on curing time.**

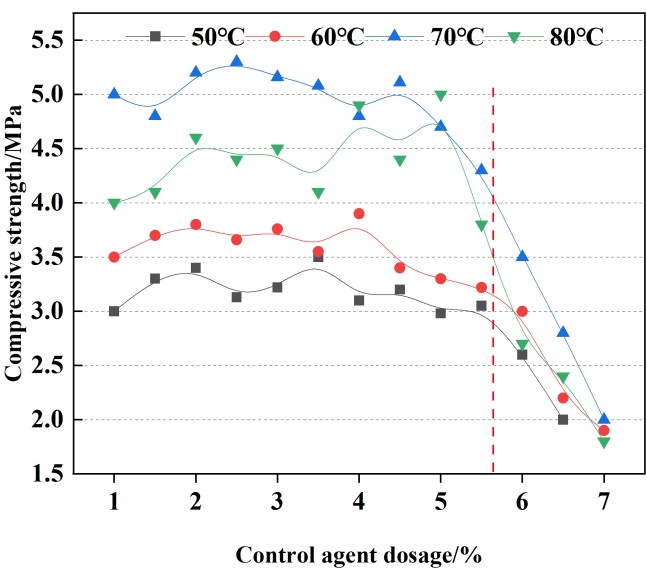

**Fig 6. Influence of temperature on compressive strength.**

exhibited a trend of first increasing and then decreasing. The sand consolidation fluid demulsified at temperatures above 60°C, while the compressive strength exceeded 3 MPa under these conditions.

The optimal formula at 80°C was determined to be 16% resin, 24% emulsifying curing agent, 5.6% control agent.

**Addition of coupling agent.** To identify a coupling agent with aqueous solubility, effective consolidation properties, and good compatibility with the control agent, KH-550 was selected as the most suitable candidate through comparative screening. To determine its optimal concentration, the curing effects of KH-550 at various mass fractions were evaluated while keeping the control agent dosage fixed at 3.6%. The coupling agent was tested within a range of 0.5% to 1.5%. As

illustrated in Fig 7, the system demonstrated strong adhesion to sand particles when the coupling agent was added at 1.0%.

## Performance evaluation of emulsion sand consolidation system

**Thermogravimetric analysis of emulsion sand consolidation system.** Thermogravimetric analysis was performed on the emulsion consolidation system, and the resulting thermogravimetric curve is presented in Fig 8. As shown in Fig 8(a), the cured epoxy resin material remained largely stable between 100°C and 250°C. From 250°C to 450°C, the epoxy resin began to decompose, undergoing rapid mass loss with a total weight reduction of 78.69%. Fig 8(b) indicates that the maximum weight loss rate of the cured product reached 8%/min at 395.1°C. A slight additional mass loss of 3% was observed between 450°C and 800°C. These results demonstrate that the epoxy resin system exhibited negligible mass loss below 200°C, indicating a stable molecular structure and good thermal resistance.

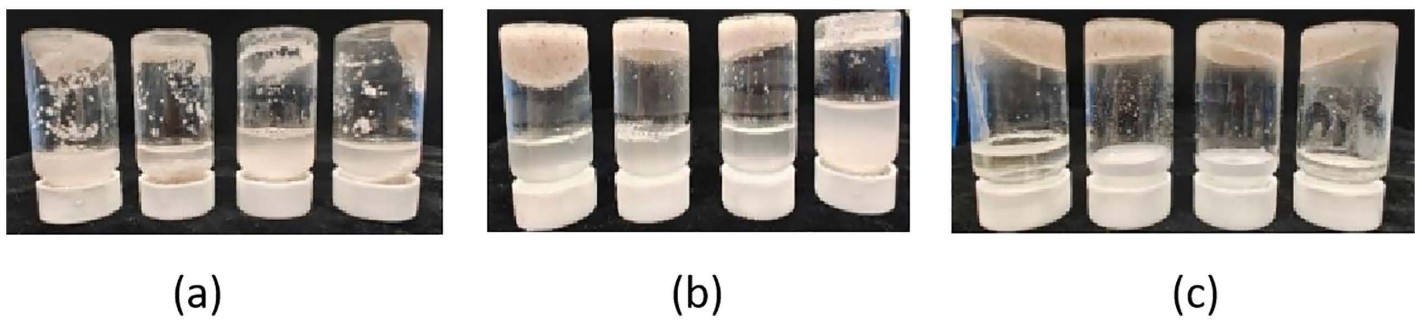

(a)  (b)  (c)

**Fig 7. Solid figure of consolidated sand column with different sand particle size.** (a) 0.5%Coupling agent addition; (b) 1.0%Coupling agent addition; (c) 1.5%Coupling agent addition.

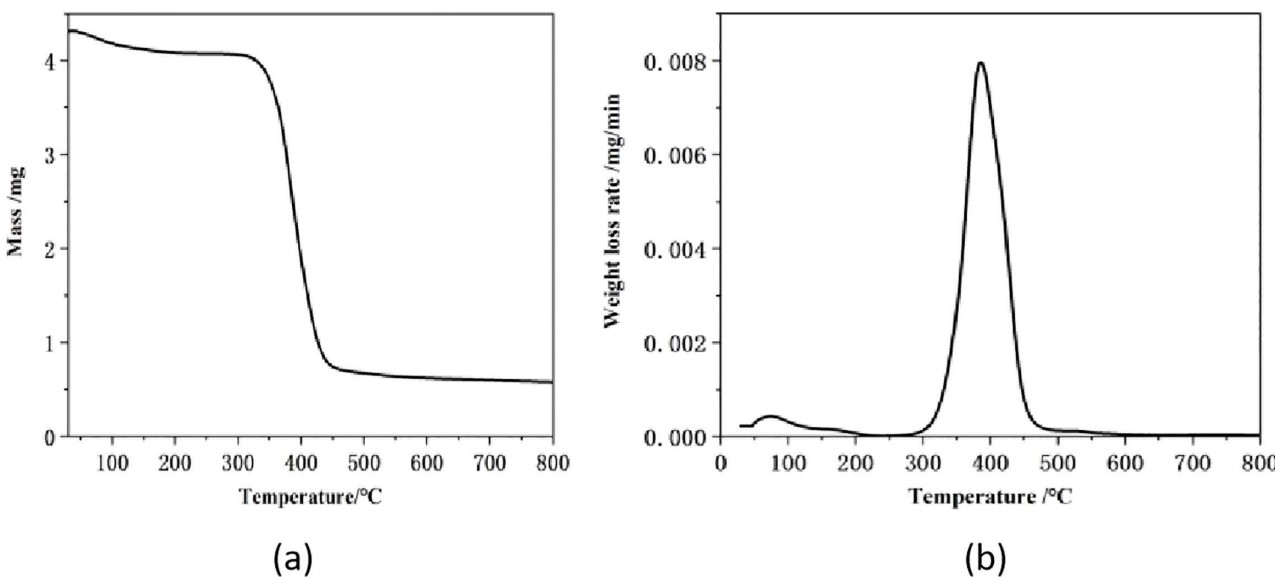

(a)  (b)

**Fig 8. Thermogravimetric analysis diagram of emulsion consolidation system.** (a) TGA curve of cured product; (b) DTG curve of cured product.

**Viscosity curve of emulsion sand system with time.**

(1)  The formation mechanism of emulsion

  Epoxy resin emulsion [32–35] can be formed either by adding an emulsifier to facilitate emulsification or through self-emulsification of epoxy resin modified with hydrophilic groups. This process disperses the epoxy resin as fine droplets (or particles) in water, resulting in a stable aqueous emulsion system. As illustrated in Figs 9 and 10, the emulsifying curing agent contains a lipophilic end, a hydrophilic end, and reactive functional groups. This amphiphilic structure allows the emulsifier to interact with epoxy resin, forming a stable aqueous emulsion. The dispersed phase of the resulting emulsion consists of emulsifier molecules on the surface enclosing the hydrophobic epoxy resin within. Simultaneously, the amine functional groups in the emulsifying curing agent can react with the epoxy resin, enabling curing during the emulsification process.

(2)  Viscosity curve of emulsion sand system

To evaluate the pumpability of the emulsion-based sand consolidation system, the emulsion was formulated with 16% resin, 24% emulsified curing agent, 5.6% control agent, 1% KH550, 1.5% modified aluminum sol, and seawater as the aqueous phase. The stability of the emulsion was monitored under both ambient temperature (25°C) and elevated temperature (80°C) conditions. Key performance indicators, including demulsification time and initial phase separation, were recorded. The viscosity of the sand consolidation fluid at different temperatures was measured using a DVNext rotational viscometer.

  As shown in Fig 11, the sand consolidation fluid prepared with seawater remained stable without phase separation after standing at room temperature for 4 days, indicating good short-term storage stability under ambient conditions. Figs 12 and 13 present the viscosity curves of the emulsion sand consolidation system over time at 80°C. It can be observed from

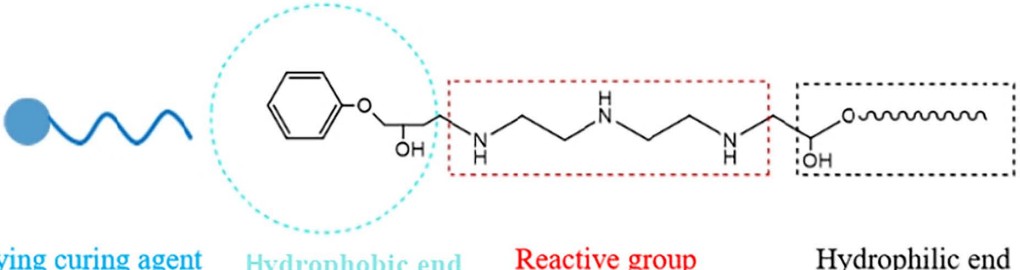

**Fig 9.  Structural formula of emulsion curing agent.**

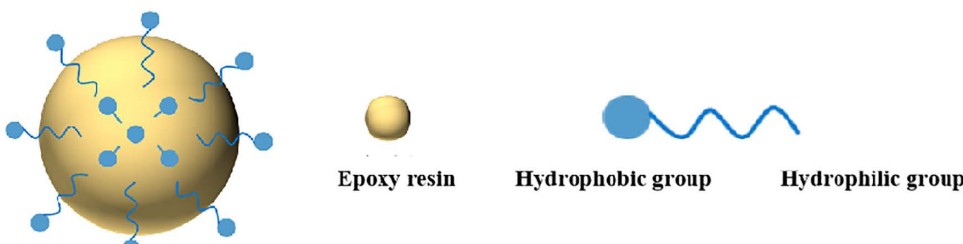

**Fig 10.  Emulsion microscopic model diagram.**

 that at the 14-hour mark, the viscosity of the system increased rapidly and substantially, rising from 8 mPa·s to 31 mPa·s within half an hour, at which point the system lost its fluidity. This indicates that curing initiated during this period, accompanied by a fast reaction rate that led to a sharp rise in the system's viscosity.

### Consolidation performance evaluation.

(1) Sand consolidation mechanism of emulsion

The emulsion sand consolidation system is injected into the target sand formation using a specialized process. During injection, the emulsion saturates the pore space completely, forming a uniform and dense consolidated matrix. The silane coupling agent plays a critical role on the sand surface, where it chemically bonds to the sand grains, significantly enhancing inter-particle adhesion. Simultaneously, emulsified resin particles form a continuous film around the sand grains, assisted by the bridging action of the coupling agent [36,30]. The thickness and strength of this film are influenced by the chemical affinity between the coupling agent and the sand particles, as well as the type of emulsifier employed. However, under elevated temperatures, the emulsifying capability of the curing agent may be compromised. Increased thermal energy can disrupt the previously stable emulsion, leading to demulsification. This breakdown adversely affects sand consolidation, as the curing agent can no longer effectively adhere to the sand surface, resulting in reduced structural

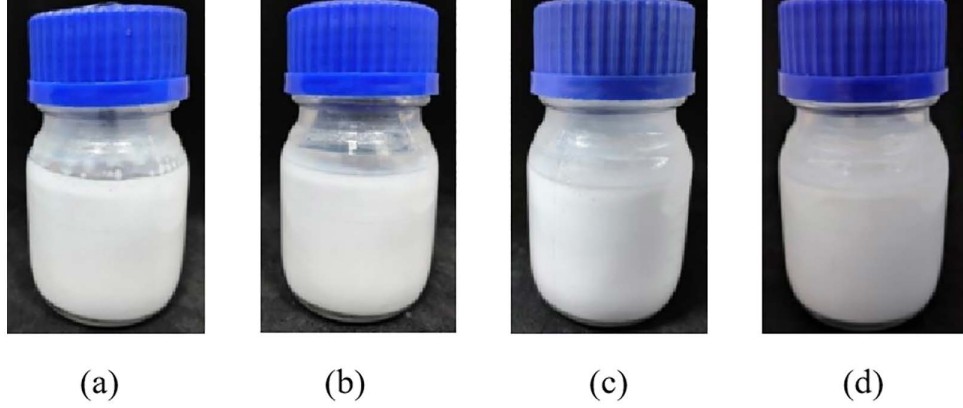

(a)     (b)     (c)     (d)

**Fig 11. Emulsion appearance at 25 °C (a) 1 d; (b) 2 d; (c) 3 d; (d) 4 d.**

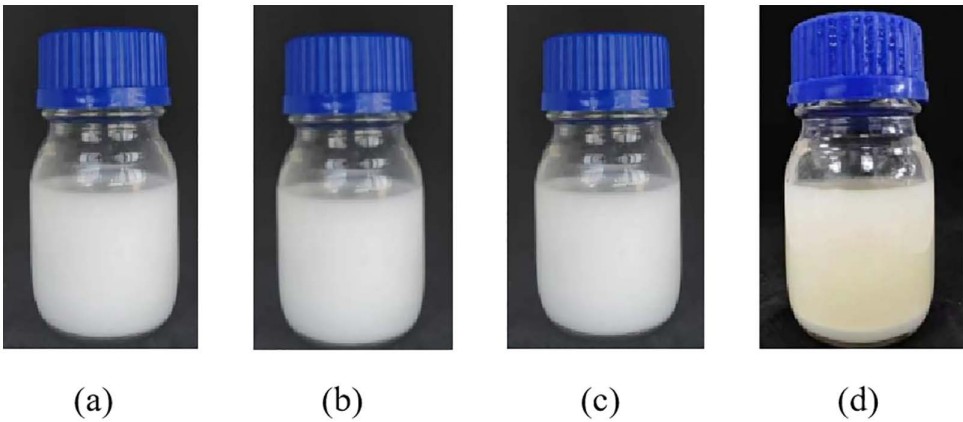

(a)     (b)     (c)     (d)

**Fig 12. Emulsion appearance at 80 °C (a) 0 h; (b) 6 h; (c) 12 h; (d) 18 h.**

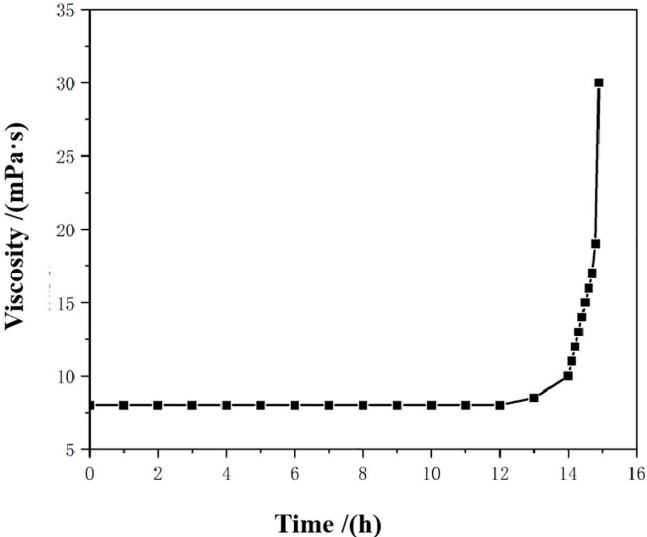

**Fig 13. Curve of viscosity change of emulsion sand system with time (at 80°C).**

integrity of the consolidated sand. Due to adsorption effects, the emulsified resin accumulates predominantly around the sand particles, trapping water within the seepage channels between grains. This mechanism not only helps maintain a high permeability retention rate but also enhances the stability and durability of the consolidated sand structure. The sand consolidation mechanism is illustrated in Fig 14.

The interaction between the sand and emulsion was further examined using the following experimental procedure: a mixture of sand and the emulsion consolidation system (containing 5% control agent) was prepared on a glass slide. The slide was then placed in a constant-temperature water bath maintained at 80 °C. At designated time intervals, the sample was retrieved and examined under a microscope. The results are presented in Fig 15.

The interaction process between the emulsion and sand was divided into five distinct stages:

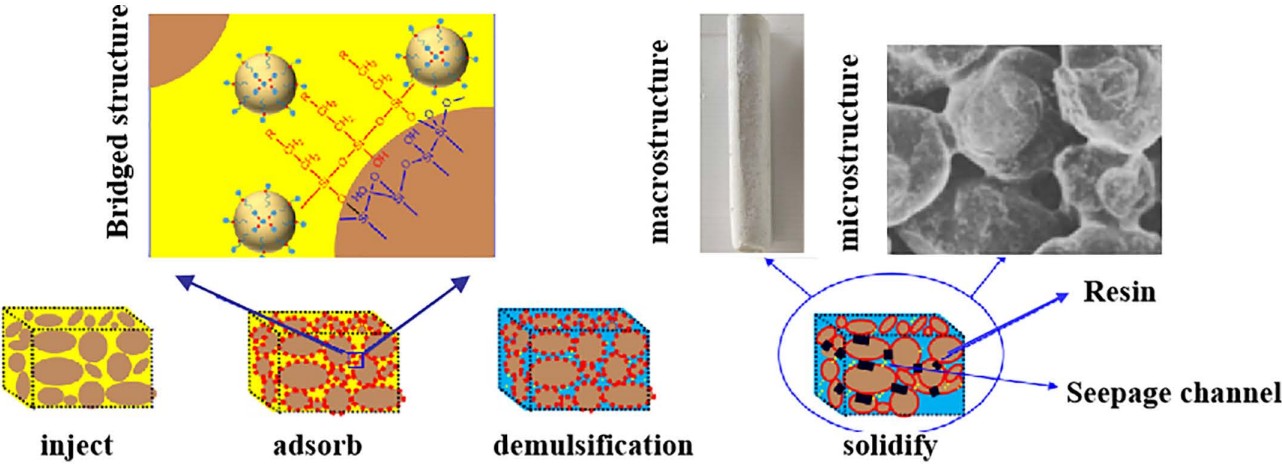

**Fig 14. Emulsion-sand interaction.**

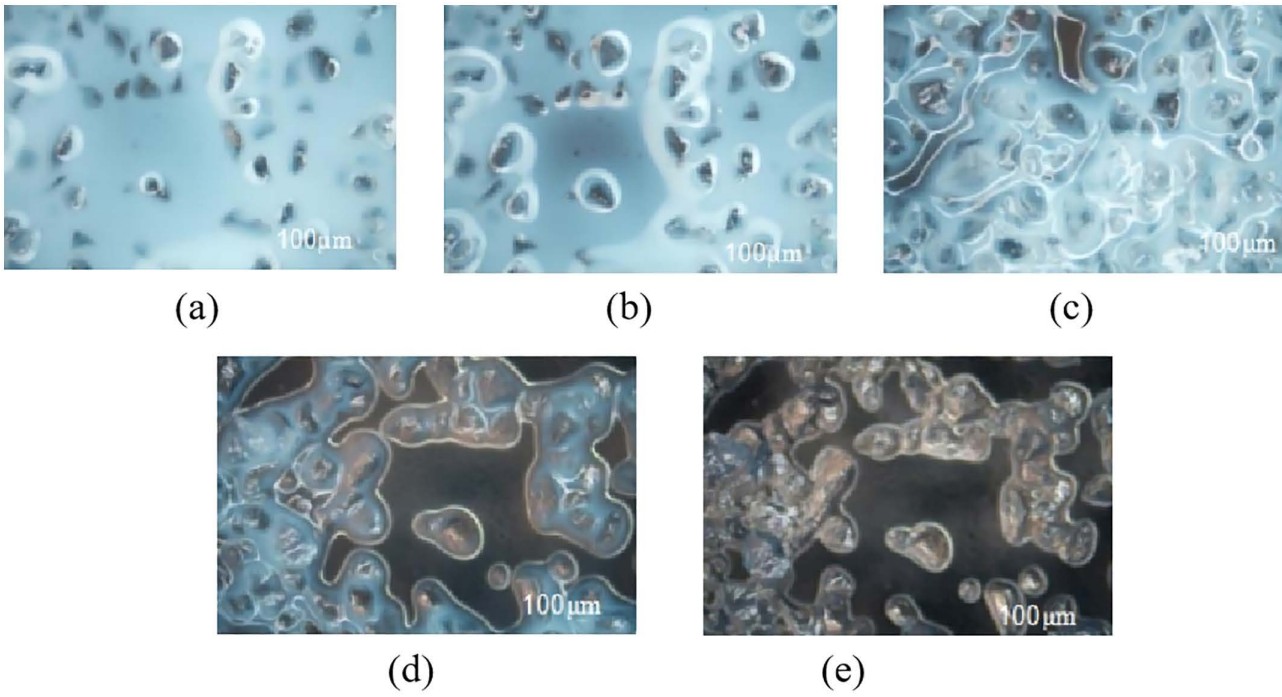

**Fig 15. Process of emulsion sand consolidation.**

(a)wetting of the sand by the emulsion;

(b)adsorption of the dispersed phase around the sand particles;

(c)onset of emulsion breaking;

(d)complete demulsification and curing;

(e)formation of consolidated sand.

The adsorption of the dispersed phase onto the sand particles after demulsification confirms a strong affinity between $SiO_2$ and the emulsion consolidation system.

(2)     Evaluation of the micro-properties of the consolidated body

    Sand consolidation fluid was injected into a sand-packed tube containing rock cuttings. The tube was then aged in an oven at 80°C for 3 days, after which the permeability after curing and compressive strength were measured. The results are shown in Table 2. As shown in Fig 16, the cross-sectional microstructure of the sand column was examined using a metallographic microscope. The analysis revealed that the resin-emulsion consolidated core contains abundant pore channels with high interconnectivity. Further SEM observations (Fig 17) indicated that the surface of the consolidated sand grains was coated with cured sand-consolidation agent, which contributes to the enhanced mechanical strength of the sand column. Moreover, the well-developed pore structure supports a high permeability retention rate.

    **Dielectric resistance.**  Consolidated cores prepared with a mass fraction of 16% resin, 24% emulsifying agent, 5.6% controller, 1% coupling agent, and 1.5% aluminum sol were immersed in various media—including formation water, 10% hydrochloric acid, and kerosene—and aged at 80 °C for 30 days, as illustrated in Figs 18 and 19. The performance parameters of the sand columns were evaluated before and after aging, with results summarized in Table 3. The data

**Table 2. Consolidation performance table.**

| ID | Permeability ($10^{-3}\mu m^2$) | | Compressive strength/MPa | | | Permeability retention rate/% |
|----|-----------|-------------|-----------|-------------|----------|---------------------------------|
|    | Inception | After curing | Entry end | Middle part | Exit end |                                 |
| 1  | 175.6     | 139.6       | 3.54      | 3.58        | 3.60     | 79.5                            |
| 2  | 259.7     | 209.2       | 3.14      | 3.24        | 3.12     | 81.5                            |

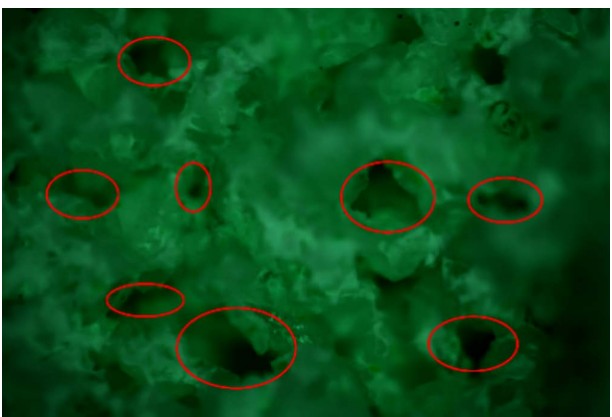

**Fig 16. Pore distribution of consolidated core.**

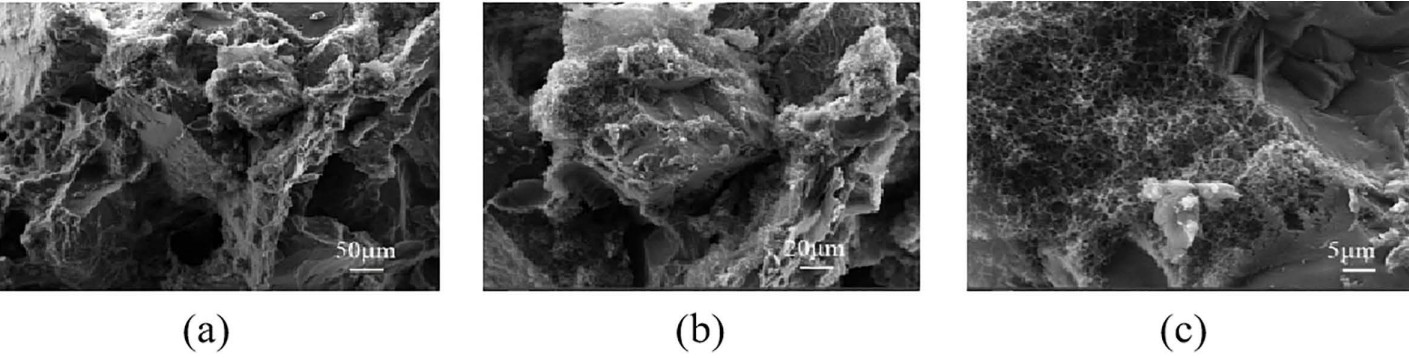

**Fig 17. Scanning electron microscopy (SEM) of the inlet end of the consolidated core.**

indicate that the compressive strength, permeability, and sand production rate of the consolidated samples showed no significant changes under prolonged exposure to different media, demonstrating excellent long-term stability and chemical resistance of the emulsion-based consolidated sand.

## Conclusion

In this study, a novel epoxy resin emulsion sand consolidation agent was developed using epoxy resin E-51 as the dispersed phase, a non-emulsifying curing agent, a coupling agent, a control agent, and high-salinity brine. To assess its field applicability, the curing time of the emulsion consolidation system, along with the compressive strength and permeability retention of the consolidated sand columns, were evaluated through laboratory experiments. The main findings are as follows:

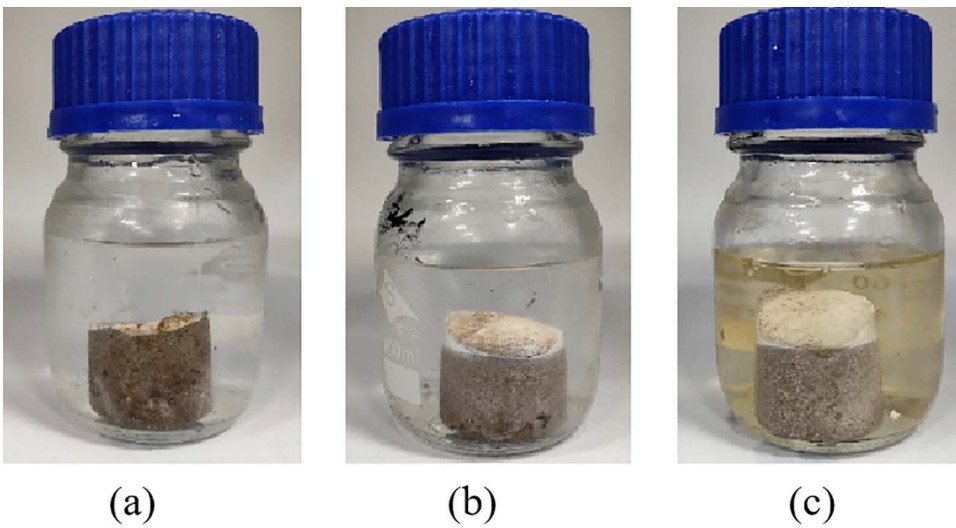

**Fig 18. Medium soaked for 0d.** (a)Formation water immersion; (b)10% hydrochloric acid immersion; (c) Kerosene soaking.

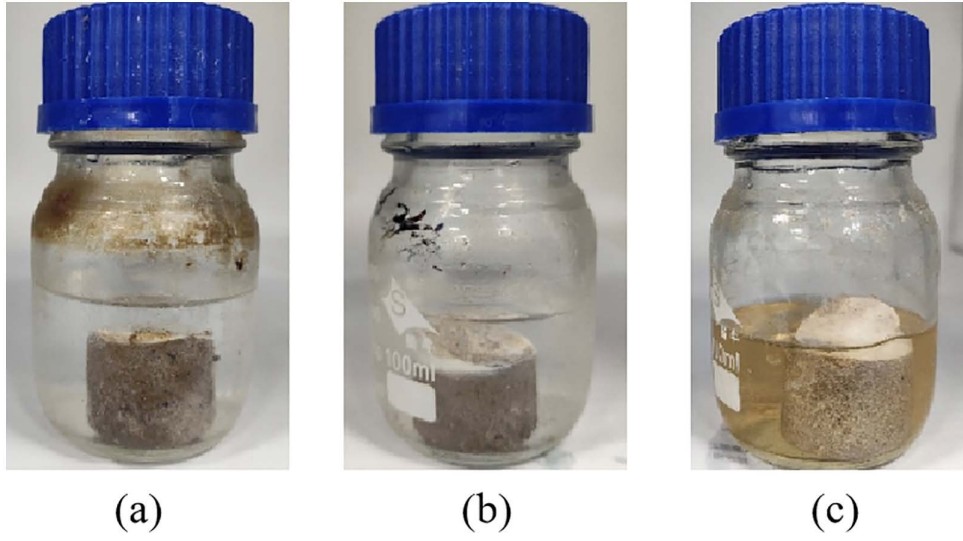

**Fig 19. Medium soaked for 30d.** (a)Formation water immersion; (b)10% hydrochloric acid immersion; (c) Kerosene soaking.

**Table 3. Evaluation of dielectric resistance.**

| Soaking medium | Soaking condition | Compressive strength/MPa | Fluid permeability/mD | Sand production rate |
|---|---|---|---|---|
| unsoaked | —— | 3.2 | 280 | 0.074 |
| kerosene | 80°C×30d | 3.1 | 265 | 0.084 |
| Formation water | 80°C×30d | 3.3 | 273 | 0.065 |
| 10% hydrochloric acid | 80°C×30d | 2.8 | 300 | 0.14 |

(1) When the dilution water has a salinity of $3.8 \times 10^4$ mg/L, the optimal formulation consists of 16% resin, 24% emulsifying curing agent, 5.6% control agent, and 1% coupling agent. Under conditions of 80°C, the resulting consolidated cores exhibited a compressive strength greater than 3 MPa, a curing time exceeding 12 hours, and a permeability retention rate above 75%.

(2) Microscopic examination of the sand column cross-section revealed a well-developed porous structure. SEM analysis further confirmed the presence of abundant and well-connected pores within the consolidated sand body. The sand grains were firmly bonded together by the cured resin.

(3) FTIR analysis clearly indicated a chemical reaction between the resin and the emulsifying curing agent, demonstrating the effectiveness of the resin emulsion system in consolidating sand particles. Thermogravimetric analysis showed minimal mass loss of the consolidated material at temperatures up to 250°C, confirming its thermal stability.

## Supporting information

**S1 Text. Effect of temperature on compressive strength.**
(XLSX)

**S2 Text. Effect of temperature on the curing time.**
(XLSX)

## Author contributions

**Conceptualization:** Xiaoxia Shang.

**Formal analysis:** Yue Pan, Ke Li.

**Funding acquisition:** Chenyang Tang.

**Investigation:** Shaobing Cai.

**Methodology:** Shaobing Cai, Juanzhe Jiang.

**Project administration:** Shaobing Cai, Juanzhe Jiang.

**Resources:** Wei Zheng.

**Software:** Wei Zheng, Ke Li.

**Supervision:** Yufei He, Yue Pan, Ke Li.

**Validation:** Yufei He, Yue Pan, Ke Li.

**Writing – original draft:** Xiaoxia Shang.

**Writing – review & editing:** Chenyang Tang.

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
