## [Decision Letter · Decision Letter 0]

18 Sep 2025

Dear Dr. Shang,

Thank you for submitting your manuscript to PLOS ONE. After careful consideration, we feel that it has merit but does not fully meet PLOS ONE’s publication criteria as it currently stands. Therefore, we invite you to submit a revised version of the manuscript that addresses the points raised during the review process.

We look forward to receiving your revised manuscript.

Kind regards,

S. Shanmugan, PhD

Academic Editor

PLOS ONE

Journal Requirements:

5. We note that you have indicated that there are restrictions to data sharing for this study. PLOS only allows data to be available upon request if there are legal or ethical restrictions on sharing data publicly. For more information on unacceptable data access restrictions, please see http://journals.plos.org/plosone/s/data-availability#loc-unacceptable-data-access-restrictions.

Additional Editor Comments :

Reviewer #1: Preparation and Evaluation of A New Sand Consolidation Agent: Emulsified Epoxy Resin

In this article titled Preparation and Evaluation of A New Sand Consolidation Agent: Emulsified Epoxy Resin,” the authors presented a study for development of new type of emulsified epoxy resin with a nonionic emulsified curing agent, fatty amine poly(epoxy ethyl ether), which is able to minimize the effect of salinity. The structure of the emulsified resin and the stability of the curing body were tested, and the effects of curing temperature, time, quartz sand mesh and control agent dosage on compressive strength and permeability were studied. This study seems to be a useful contribution to the literature, and it can be accepted after addressing the following minor issues

Reviewers' comments:

Reviewer's Responses to Questions

**Comments to the Author**

1. Is the manuscript technically sound, and do the data support the conclusions?

Reviewer #1: Yes

2. Has the statistical analysis been performed appropriately and rigorously?

Reviewer #1: Yes

3. Have the authors made all data underlying the findings in their manuscript fully available?

Reviewer #1: Yes

4. Is the manuscript presented in an intelligible fashion and written in standard English?

Reviewer #1: Yes

Reviewer #1: Article Review

Preparation and Evaluation of A New Sand Consolidation Agent: Emulsified Epoxy Resin

In this article titled Preparation and Evaluation of A New Sand Consolidation Agent: Emulsified Epoxy Resin,” the authors presented a study for development of new type of emulsified epoxy resin with a nonionic emulsified curing agent, fatty amine poly(epoxy ethyl ether), which is able to minimize the effect of salinity. The structure of the emulsified resin and the stability of the curing body were tested, and the effects of curing temperature, time, quartz sand mesh and control agent dosage on compressive strength and permeability were studied. This study seems to be a useful contribution to the literature, and it can be accepted after addressing the following minor issues:

1. The abstract should be improved to enhance flow and to provide the literature gap and the importance of this study. It should be more self-explanatory, especially about the major types of architecture and their relative strengths and weaknesses. It should introduce the field at least in one line and mention the research gap to establish the novelty of this study.

2. The Keywords should be revised to include more relevant keywords and should not include words already mentioned in the title to improve the searchability of the article. For better Search Engine Optimization and searchability, more focused keywords should be included while omitting the generic ones.

3. The English language in this manuscript should be improved. It should also be more elaborate.

4. The methodology section need more explanation.

5. The article should include more relevant literature; it currently consists of very few references to successfully establish the need for the current study.

6. The motivation behind this study should be included with more detail, it should be in proper flow, introducing the gap and research question.

7. The suppositions and boundary conditions should be explicitly declared.

8. The overall flow of language is incoherent, and there are various grammatical and spelling mistakes. These issues should be resolved.

The End

**Do you want your identity to be public for this peer review?** For information about this choice, including consent withdrawal, please see our Privacy Policy

Reviewer #1: **Yes: ** Ali Raza Shafqat

---

## [Author Response · Author response to Decision Letter 1]

23 Sep 2025

Dear Dr. Ali Raza Shafqat,

Thank you for giving us the opportunity to revise our manuscript entitled“ Seawater-Resistant Emulsified Epoxy Resin for Effective Sand Control In Unconsolidated Sandstone Oil Reservoir” (ID: PONE-D-25-36037). We are also grateful to the reviewers for their time, effort, and insightful comments, which have helped us significantly improve the quality of our paper. We have carefully considered all the comments and have made extensive revisions to the manuscript accordingly. The point-by-point responses to the reviewers’ comments are listed below.

Reviewer’s Comments:

Comment 1: 

The abstract should be improved to enhance flow and to provide the literature gap and the importance of this study. It should be more self-explanatory, especially about the major types of architecture and their relative strengths and weaknesses. It should introduce the field at least in one line and mention the research gap to establish the novelty of this study.

Response: We sincerely thank the reviewer for this suggestion. We have thoroughly revised the abstract to improve its flow and clarity. Specifically, we have: Added an introductory sentence to establish the research field. Clearly stated the research gap and the motivation for this study. Provided a concise explanation of the major architectural types discussed, along with their relative strengths and weaknesses. Emphasized the novelty and importance of our work in addressing the identified gap. We believe the revised abstract is now more comprehensive and self-explanatory.

“Abstract Sand production in oil wells is recognized as a persistent challenge during oilfield development, adversely affecting well productivity and operational stability. Chemical sand control methods, particularly resin-based sand consolidation, are considered a promising solution due to their operational simplicity and effectiveness. However, conventional emulsified resins are known to be highly sensitive to high-salinity environments, which can lead to emulsion destabilization and reduced consolidation strength. To address this limitation, a novel emulsified epoxy resin system was developed in this study using a nonionic emulsifying curing agent—fatty amine poly(epoxy ethyl ether)—by which salinity tolerance is significantly enhanced, supporting dilution water salinity up to 3.8×10⁴ mg/L. Through single-factor experiments, an optimal formulation was identified as 16% epoxy resin, 24% emulsified curing agent, 1% coupling agent, and 5.6% stabilizer. The molecular structure of the emulsified resin and the stability of the cured matrix were thoroughly characterized. The effects of curing temperature, time, sand particle size, and stabilizer dosage on compressive strength and permeability were systematically evaluated. It was demonstrated that after being cured at 80 °C for 12 hours, the consolidated cores achieved a compressive strength exceeding 3 MPa with permeability retention above 75%. Furthermore, the consolidated cores were shown to exhibit excellent long-term stability, maintaining their mechanical and flow properties after 30-day immersion in kerosene, 10% HCl, and formation water. This study bridges a critical research gap in high-salinity applications of water-based resin emulsions and provides a robust technical solution for sand control in challenging reservoir environments.”

Comment 2: 

The Keywords should be revised to include more relevant keywords and should not include words already mentioned in the title to improve the searchability of the article. For better Search Engine Optimization and searchability, more focused keywords should be included while omitting the generic ones.

Response: We appreciate this helpful suggestion. We have revised the keywords section by:

Removing words that already appear in the manuscript title. Adding more specific and relevant keywords that accurately represent the core content of our study and are likely to be used in literature searches. Ensuring the new keywords are focused and appropriate for improving the article's discoverability.

“Keywords chemical sand control; nonionic emulsifier; high-salinity tolerance; permeability retention; long-term stability; sand fixing properties”

Comment 3: 

The English language in this manuscript should be improved. It should also be more elaborate.

Response: We thank the reviewer for pointing this out. The manuscript has been extensively revised to improve the English language and clarity. We have carefully polished the language throughout the paper, corrected grammatical errors, and elaborated on key points to enhance readability and comprehensiveness.

Comment 4: 

The methodology section need more explanation.

Response: We agree with the reviewer that a more detailed methodology was needed. We have significantly expanded the methodology section to provide a clearer, step-by-step explanation of the research design, data sources, analytical procedures, and computational methods employed. This added detail will ensure the study is reproducible and its rigor is clearly communicated. The modifications can be found in the manuscript from line 131 to line 203.

Comment 5: 

The article should include more relevant literature; it currently consists of very few references to successfully establish the need for the current study.

Response: We thank the reviewer for this valuable comment. We have conducted a thorough review of the recent and relevant literature and have added numerous citations throughout the introduction and background sections. These new references better contextualize our study within the existing body of knowledge and more effectively establish the research gap that our work aims to address. The modifications can be found in the manuscript from line 37 to line 110.

Comment 6: 

The motivation behind this study should be included with more detail, it should be in proper flow, introducing the gap and research question.

Response: We appreciate this suggestion. The introduction has been restructured and rewritten to provide a more logical narrative flow. We have elaborated on the motivation for the study, clearly leading the reader from the general background to the specific research gap, and then to the research questions and objectives that our study tackles.

Comment 7: 

The suppositions and boundary conditions should be explicitly declared.

Response: We thank the reviewer for this important feedback. We have now explicitly stated the key assumptions and boundary conditions underlying our research in a dedicated subsection within the methodology. This clarification ensures transparency and helps readers understand the scope and limitations of our study. The modifications can be found in the manuscript from line 188 to line 203.

“Experimental Assumptions and Boundary Conditions

The experimental investigation and subsequent analysis in this study were conducted under the following explicit assumptions and boundary conditions: Sand Grain Composition: The consolidation experiments primarily assume quartz sand as the representative formation sand. The presence of significant clay minerals or other highly reactive minerals may alter the chemical interaction and consolidation effectiveness. Formation Homogeneity: The laboratory-prepared sand packs assume a relatively homogeneous and unconsolidated formation model. Complex heterogeneities, natural fractures, or pre-existing stresses in real reservoirs are not accounted for. Curing Conditions: The curing process is assumed to be primarily thermally driven at the specified temperatures (e.g., 80°C). The potential influences of downhole pressure on curing kinetics and final properties were not investigated in this phase of study. Fluid Compatibility: The assessment of chemical resistance assumes exposure to the specified fluids (formation water, 10% HCl, kerosene) under static conditions. Dynamic flow conditions or interactions with mixed/complex wellbore fluids are beyond the current scope. Scale: The experiments are conducted on a laboratory scale. Scaling effects for field application, including placement efficiency, radial flow effects, and large-volume mixing, are not considered. These boundaries define the specific context within which the results and conclusions of this study are valid.”

Comment 8: 

The overall flow of language is incoherent, and there are various grammatical and spelling mistakes. These issues should be resolved.

Response: We sincerely apologize for these issues. We have performed a comprehensive line-by-line edit of the entire manuscript to rectify grammatical and spelling errors. Furthermore, we have restructured sentences and paragraphs to improve the overall coherence, logical flow, and readability of the text.

Once again, we thank you and the reviewers for these constructive comments. We have addressed all the points raised and believe the revisions have substantially improved the manuscript. We hope the revised version is now suitable for publication in PLOS ONE.

Sincerely,

Xiaoxia Shang

Shangxx27@163.com

---

## [Editor Report · Decision Letter 1]

17 Oct 2025

Preparation and Evaluation of A New Sand Consolidation Agent: Emulsified Epoxy Resin

PONE-D-25-36037R1

Dear Dr. Xiaoxia Shang

We’re pleased to inform you that your manuscript has been judged scientifically suitable for publication and will be formally updated Reviewer 1 comments for publication once it meets all outstanding technical requirements.

Kind regards,

S. Shanmugan, PhD

Academic Editor

PLOS ONE

Additional Editor Comments (optional):

Process

Reviewers' comments:

Follow up Reviewer 1 comments

---

## [Editor Report · Acceptance letter]

PONE-D-25-36037R1

PLOS ONE

Dear Dr. Shang,

I'm pleased to inform you that your manuscript has been deemed suitable for publication in PLOS ONE. Congratulations! Your manuscript is now being handed over to our production team.

Kind regards,

on behalf of

Dr. S. Shanmugan

Academic Editor

PLOS ONE